# Juvenile Osteochondritis Dissecans: A Case Report

**DOI:** 10.3390/diagnostics14171931

**Published:** 2024-09-01

**Authors:** Hermann Nudelman, Aba Lőrincz, Tamás Kassai, Gergő Józsa

**Affiliations:** 1Department of Thermophysiology, Institute for Translational Medicine, Medical School, University of Pécs, 12 Szigeti Street, 7624 Pécs, Hungary; nuhwaao.pte@tr.pte.hu (H.N.); aba.lorincz@gmail.com (A.L.); 2Division of Pediatric Surgery, Traumatology, Urology and Pediatric Otolaryngology, Department of Pediatrics, Medical School, University of Pécs, 7 József Attila Street, 7623 Pécs, Hungary; 3Department of Pediatric Traumatology, Péterfy Hospital, Manninger Jenő National Trauma Center, 17 Fiumei Street, 1081 Budapest, Hungary

**Keywords:** osteochondritis dissecans, OCD, mosaicplasty, elbow, talar, articular surface, congruency

## Abstract

(1) Background: This report aims to illustrate the development, progression, diagnosis, and treatment of chronically present articular surface lesions. (2) Methods: In this report, two patients are described from the point of the initial presentation of symptoms to surgical consultation based on radiologic findings. These patients underwent corrective surgery in the form of mosaicplasty to repair lesions present on the articular surface and the underlying subchondral bone. (3) Discussion: Diagnosing juvenile OCD remains challenging due to its variable clinical presentation and minute radiologic discoveries. X-rays are useful; however, the gold standard remains arthroscopy, which can be both diagnostic and therapeutic. Future prospects include the use of novel sonographic methods and the use of artificial intelligence within the given modalities. (4) Conclusions: The detailed imaging provided by MRI, combined with the insights from X-rays and potentially other modalities, allows for a nuanced understanding of this disease. This comprehensive approach ensures that treatment decisions are well-informed, optimising outcomes for young patients with this condition.

## 1. Introduction

Juvenile osteochondritis dissecans (OCD) is a condition which occurs when the blood supply is diminished in a segment of bone and its surrounding cartilage, which may necrotise and separate from the underlying tissues [1]. The condition affects primarily children and adolescents who are physically active. Towards the end of the 1800s, Franz König described the condition as an inflammatory process which yields loose bodies within the joint. However, novel research methods have revealed that the primary issue is not inflammation but is rather due to disruption of the blood supply of the subchondral bone [2]. As time passed, this shifted the attention regarding the development of OCD towards mechanical and vascular factors. The diagnostic criteria have evolved, but the recognition of juvenile OCD is still a significant challenge.

Juvenile OCD is much more prevalent than in adults, with no exact number of cases per year, only rough estimates. Most frequently, the knee (9–29/100,000), ankle (2–5/100,000), and elbow (2/100,000) are affected, with the possibility of smaller joint involvement. The condition is rare under the age of 10, with an increased risk from age 12 to 19 [1]. Generally, boys are four times more likely to suffer from OCD in the knee and the elbow; however, girls have a 1.5 times higher risk of developing the condition in the ankle. Typically, in the knee, the defect will present at the posterior-central medial condyle (63–85%), followed by the lateral condyle inferior-centrally (15–32%), the inferomedial aspect of the patella (1.5–10%), and the trochlea (2%) [3]. Bilateral involvement occurs in 7–29% of cases. In the elbow, the capitulum is affected in an anterolateral or central position (97.5%) ten times more often than the trochlea (2.5%). Talar OCD lesions are commonly posteromedial (71%), anterolateral (22%), or central (3.5%). Talar bilateral involvement is rare. OCD tends to affect active children and young adults [1,3]. Participation in high-intensity sports activities and athletics is commonly associated with developing the condition in the knee (60%), elbow (84%), and ankle (64%) [3]. The condition is frequently associated with football, basketball, baseball, and gymnastics. The risk of developing the condition further increases with obesity. Moderate obesity will increase the risk of developing knee OCD, while extreme obesity will increase the risk of talar OCD. A trend of early onset may be associated with early intensive sports activities or severe childhood obesity [3,4].

The condition’s aetiology is not well understood, even though advancements have been made regarding its pathogenesis. As the name suggests, an inflammatory process should be present; however, minimal evidence supports this, either histologically or biochemically [1,3]. Human histochemical studies have shown necrosis in the subchondral bone and the epiphyseal cartilage, cartilage degradation, and matrix metalloproteinase expression linked to chondrocyte apoptosis. The presence of osteonecrosis due to steroids, radiation, hemoglobinopathies, or chemotherapy could also potentially provide a pathway for degenerative changes, but the most recent systematic review suggests OCD’s cause likely involves both mechanical and biological factors [1,5]. The microtrauma theory was introduced in the 1950s to describe the pathogenesis of the condition, but since then, the literature has provided arguments for repetitive microtrauma and anatomic variations in the condyles, discoid meniscus, or posterior cruciate ligament [6,7,8]. Incorrect knee positioning, obesity, soft tissue instability, and lower limb misalignment have been shown to be part of the biomechanical background of developing the condition [9,10]. Genetic involvement was suggested due to the increased disease incidence in monozygotes [8,11]. Hypovitaminosis, namely vitamin D, has been proven to impact disease development significantly [12,13,14].

The susceptibility to ischemic injury can be explained by the vascularity of subchondral bone, which has been demonstrated to be supplied with poor arterial anastomoses, such as in the case of bowel mesentery. Tóth et al. used imaging techniques to demonstrate the involvement of abnormal vascular structure at the predicted sites of OCD in cadavers [2]. The subarticular epiphyseal cartilage shapes adult joint geometry, the survival of which depends on proper blood supply. As the secondary ossification centres expand during growth, these vessels regress [2,15]. The preceding condition, called osteochondritis latens, was shown to be caused by premature regression of these vessels, which led to necrosis of the epiphyseal cartilage in animal samples [5,16]. This lesion will spare the articular surface and the subchondral bone, involving only the epiphyseal cartilage. Osteochondritis manifesta results in insufficient mechanical reinforcement to the overlying cartilage. This occurs as growth continues and the ossification reaches the lesion, and a necrotic piece of cartilage will be trapped inside the subchondral bone [17]. These lesions may heal on their own, as demonstrated in animal studies [18]. Nonetheless, osteochondritis dissecans may develop from these lesions, either due to trauma or repetitive microtrauma [18].

Diagnosis is made difficult due to varying symptoms, depending on severity. Mechanical symptoms, namely locking and catching, are indicative of loose bodies or unstable lesions [1,19]. Those who live an active lifestyle or are involved in sports activities are at a higher risk, with symptoms often surfacing following repetitive microtrauma or a specific injury. Diagnostic delays can occur due to the non-specific nature of early symptoms, underscoring the importance of thorough clinical and radiological evaluation for accurate diagnosis.

The primary diagnostic tools include X-ray and MRI; however, novel sonographic methods have been described in recent years, detecting elbow injuries [20]. Many classifications exist based on either the location or severity of the injury, depending on the type of imaging. A summary of these can be viewed in Table 1. However, simplified staging can be utilised for all OCD lesions characterising the severity of the injury into the following categories: (I) subchondral defect, (II) surface defect, (III) loose body, (IV) displaced fragment and articular incongruence.

The most important factors determining the treatment plan are lesion stability and skeletal maturity. Generally, non-operative treatment is adequate in cases of stable lesions and open physis or in asymptomatic cases. This entails 3–6 months with activity modification and weight-bearing limitations with the possibility of immobilisation, which is controversial [25]. Surgical correction is indicated for unstable lesions or failed conservative management. The options for stable lesions include transarticular or retroarticular drilling [3]. Unstable but salvageable lesions can be treated by aiming to correct the fracture, realign the articular surface, and increase the blood supply. This can be achieved by headless, partially threaded compression screws, which can be metal or made out of bioabsorbable materials such as PLGA [26]. If a loose body cannot be saved due to fragmentation, necrosis or cartilage degeneration restorative techniques should be applied. The options include osteochondral autograft transplant—also known as mosaicplasty—autologous chondrocyte implantation, or allograft transplants, as loose body excision alone leads to poor outcomes [27,28,29]. Mosaicplasty has proven to be effective for paediatric OCD through several studies [28,30,31]. A key advantage lies in restoring the articular surface mimicking the native cartilage. Another advantage of this method when utilised in the paediatric population, is this population’s high capacity for healing and integration. Their highly vascularised structures provide better regenerative potential than those of adults. Autografts further minimise the risk of graft rejection and the need for a donor [28,30].

## 2. Materials and Methods

Patient 1: The first patient arrived at the clinic in October of 2021 at the age of 12 with persistent pain which resulted from an injury a month prior. At the initial evaluation, an X-ray was performed which could not identify a fracture; however, it raised the possibility of OCD (Figure 1).

With proper conservative treatment, the pain persisted throughout the recovery period. Due to suspicion of osteochondral injury, an MRI was requested, which revealed the unstable defect of the talar surface, medially (Figure 2).

Due to symptomatic disease progression, surgery was advocated for, which took place on 2023.06.14; the patient was aged 15 at this point. Under general anaesthesia, an osteotomy was performed to access the talar surface giving a view of the defect (Figure 3).

The OCD lesion was removed and mosaicplasty was carried out. The graft piece was obtained from the non-weight-bearing surface of the left femoral lateral condyle. After proper positioning and adequate results of the articular surface and its congruency, the osteotomy was repaired and fixed with two absorbable nails made out of poly-L-lactide-co-glycolic acid, with a composition of 85G:15L (PLGA) (Figure 4). After closure, swathing and bandaging took place and the ankle was fixed with an ankle brace.

In the postoperative period, no complications were noted, and radiologic control examinations confirmed the proper position of the mosaicplasty as well as the osteotomy (Figure 5). Control examinations took place 2 months after the operation and were unremarkable (Figure 6).

The child underwent physiotherapeutic training in the postoperative period. The ankle was immobilised, and no body weight bearing was permitted for the initial 4 weeks. The knee was moved with continuous passive motion exercises (CPM), as was the ankle, until the pain threshold was reached during movement. Mobilisation could occur with the help of clutches, and further examinations are planned as part of the continued follow-ups. The one-month post-op control evaluation concluded that the implants and the graft were well in place, and no complaints or complications were reported during the healing period. Further long-term follow-up is currently ongoing.

Patient 2: This child was an athlete and performed continuous active movement in the form of gymnastics on a daily basis. Her complaints started half a year prior to December of 2021, at the age of 14, when they were presented at the clinic. The patient did not remember a specific injury which might have elicited the original onset; however, she complained of several minor injuries and falls which occur with regular frequency. The ROM of the left elbow had decreased to 20–100 degrees. Neurovascular pathologies could not be identified or elicited with specific tests. While ultrasound has been demonstrated to detect and evaluate OCD lesions well, particularly in the humeral capitellum, it has limitations in visualising deeper structures and providing detailed information on the extent of cartilage involvement. Ultrasound has been proposed as a useful method in the literature, but it is not routinely performed in our clinic. The lack of hands-on training and limited experience with the technique among our staff led to the utilization of other diagnostic approaches, as further training would be required to perform it accurately. This made MRI the preferred imaging modality for this case. An MR examination took place which revealed a loose chondral body (Figure 7).

On 2022.01.07, under general anaesthesia, the patient underwent surgery. The joint was accessed through a minimally invasive anterolateral incision (about 3 cm) (Figure 8A). After visualisation, the loose chondral fragment was removed, as was some excess coagulum originating from the synovium (Figure 8C).

The defect was 10 mm × 11 mm in size and was corrected by mosaicplasty. The graft piece was obtained from the non-weight-bearing surface of the femur’s lateral condyle and was attached to yield a fresh congruent articular surface (Figure 9). The joint capsule was reconstructed and the wound was closed with 4-0 Vycryl absorbable thread with a continuous intracutaneous technique. Before swathing and bandaging, a nerve block was initiated with a sodium-channel blocker and Tonogen to lessen the immediate post-op pain.

After congruence was confirmed, the joint capsule was reconstructed and the incision was sutured with 4-0 Vycryl thread and a continuous intracutaneous suture. Following swathing and bandaging, an elbow brace was applied in the 90-degree position. A control X-ray after surgery confirmed the proper congruence (Figure 10A,B).

The child underwent physiotherapeutic training in the postoperative period. The elbow was immobilised for the first 4 weeks. The elbow was trained with continuous passive motion exercises (CPM) until the pain threshold was reached during movement. During the first 8 weeks, elbow extension–flexion was limited to 20–120, meaning that extension and flexion were incomplete. The asymmetry between the two extremities can be seen in several different positions (Figure 11). The incongruence was corrected with the help of one year of physiotherapy. The patient gradually returned to sports and gymnastics after one year.

The control MR imaging at six months proved that the lesion was corrected adequately and that the fixation maintained the function and congruency of the joint (Figure 12). However, effectiveness and long-term success should be further evaluated in the patient during adolescence and young adulthood.

After one year, the ROM was complete and could yield 0–130 degrees of movement (Figure 11A-2–D-2). Pronation and supination were painless and no complications or complaints were noted even with physical activity.

## 3. Discussion

Variable clinical presentation and subtle radiological findings make the diagnosis of juvenile OCD still challenging. Comprehensive imaging techniques are essential for the proper staging of the condition and for determining the appropriate treatment strategy. The variability in symptom presentation often leads to delayed diagnosis or misdiagnosis, which may complicate outcomes.

The primary diagnostic modalities for OCD include X-ray and MRI, with each offering distinct advantages. X-rays, being the initial imaging technique, are valuable for identifying gross abnormalities such as lesions or bone fragments. However, their sensitivity is limited, particularly in the early stages of OCD when subchondral bone changes are minimal or absent. In the cases presented, X-rays were initially inconclusive, highlighting the diagnostic challenge and underscoring the importance of more advanced imaging techniques. MRI, on the other hand, provides a detailed view of both bone and soft tissue, making it the gold standard for diagnosing OCD. It is particularly effective in detecting early subchondral changes, the presence of loose bodies, and cartilage status, which are not visible on X-rays. The use of MRI allows for a more nuanced classification of lesions, aiding in the assessment of lesion stability and the extent of cartilage involvement. This is crucial for planning treatment, as the choice between conservative management and surgical intervention often hinges on these details. MRI findings such as the presence of high signal changes behind the lesion, indicating fluid or fibrous tissue, can signal instability, thus guiding more aggressive treatment approaches. In today’s world, AI programming could also aid in the diagnosis of OCD and may be expected to routinely check X-ray images for defects within the near future, as it has been used for several conditions [32,33,34].

Accurate staging through radiological evaluation not only helps in the initial diagnosis but also in monitoring disease progression and the response to treatment. For instance, conservative management, often preferred in stable lesions, can be monitored for healing or progression through follow-up MRIs. Conversely, surgical intervention, including techniques like mosaicplasty, is considered when imaging shows unstable lesions or significant cartilage damage [31]. Generally, follow-up should take place for one year at minimum, but optimally, it should last for 3–5 years. MRI should be conducted yearly to properly monitor healing or disease progression to avoid long-term complications. Graft rejection, infection, and bleeding are the major early complications, while later on, joint stiffness, arthritis, graft necrosis, or donor site morbidity may occur [27].

Arthroscopy can be both therapeutic and diagnostic, but this procedure requires experience. This should take place if the patient has persistent pain, a deficit in the range of motion, and locking or catching [22]. If the arthroscopic findings can support the radiological results, a therapeutic approach should take place with debridement, removal if necessary, and restorative approaches, such as mosaicplasty [19].

Ultrasound has been proven effective as a diagnostic tool for the evaluation of OCD of the talus or the humeral capitellum [35]. It offers a non-invasive, radiation-free method to assess the integrity of the articular cartilage. Cartilage irregularities, subchondral bone defects, and loose bodies can be detected, as can effusion, and the dynamic assessment of lesion stability provides an additional benefit in contrast to static modalities such as X-rays. However, just as in the case of the talus, the effectiveness of ultrasound diagnosis for the humeral capitellum largely depends on the skill and experience of the operator. In cases where the lesion is situated deeply or the joint holds complex anatomy, ultrasound may not provide as clear or comprehensive an assessment as MRI [20,35]. Additionally, the literature is still lacking in standardised protocols and the limited routine use of this diagnostic method further restricts its utility. In our cases, we opted for MRI to ensure a thorough evaluation. However, ultrasound remains a valuable approach, particularly when advanced diagnostics are not available or when quick dynamic assessments are needed [35].

The primary reason for selecting mosaicplasty as a treatment method was the size and nature of the lesion. In these cases, the defects were large and unstable, posing a significant risk for the progression of degenerative joint disease if left untreated [36]. This method allowed us to provide immediate structural support and joint surface restoration. Secondarily, the patients’ young ages and active lifestyles prompted the treatments, as the restoration of a smooth, durable joint surface was needed to allow them to return to sports activities without long-term consequences [37]. Lastly, studies have shown that significant improvements in joint function can be achieved, as can a decreased chance of long-term complications, compared to other methods [37,38].

The limitations of this study primarily stem from its retrospective nature and the small sample size, as it contains only two case reports. This restricts the generalisability of the findings to a broader population. Additionally, this study focuses on specific imaging and surgical interventions which may not encompass the whole spectrum of treatment options for the condition. Furthermore, while this study provides insight into the treatment outcome of mosaicplasty, it lacks long-term follow-up data, which would be necessary to fully assess the efficacy and potential complications of the procedure over time. Another limitation of this study is the reliance on conventional imaging techniques; newer imaging modalities which might offer more detailed evaluations were not explored and should be further evaluated. Lastly, this study does not incorporate a control group, which limits our ability to compare outcomes across different treatment modalities or to assess the natural course of the disease without significant intervention. Future studies with larger sample sizes, longer follow-up periods, and the inclusion of advanced imaging techniques and control groups would help to overcome these limitations and provide more comprehensive data on the management of juvenile OCD.

## 4. Conclusions

In conclusion, the integration of comprehensive diagnostic and radiological evaluation is indispensable in the management of juvenile OCD. The detailed imaging provided by MRI, combined with the insights from X-rays and potentially other modalities, allows for a better understanding of the disease. Further large-scale studies are still required to establish a highly sensitive and specific approach. Mosaicplasty remains a fine choice for severe, large, or displaced lesions, with promising outcomes.

## Figures and Tables

**Figure 1 diagnostics-14-01931-f001:**
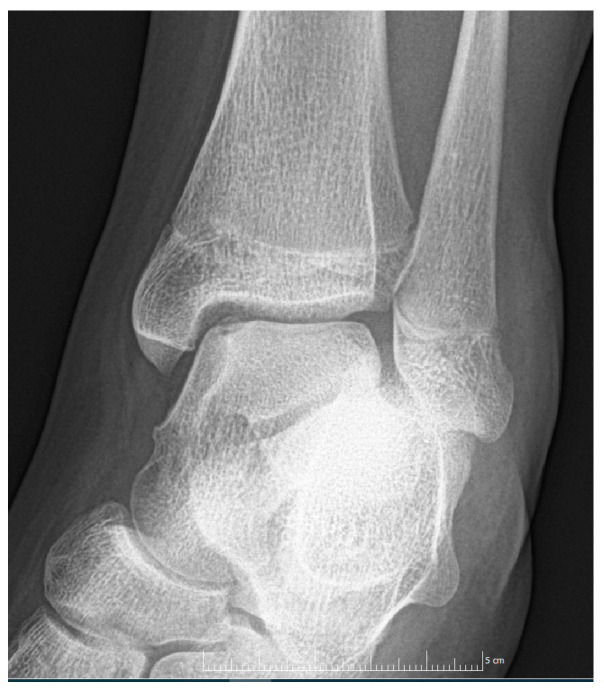
The first X-ray taken after the initial injury.

**Figure 2 diagnostics-14-01931-f002:**
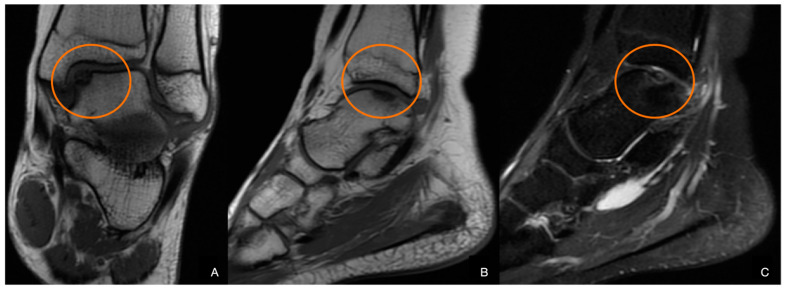
MR Images: A-P (**A**), side view (**B**), and T2-weighted (**C**). The defect is highlighted in the red circle.

**Figure 3 diagnostics-14-01931-f003:**
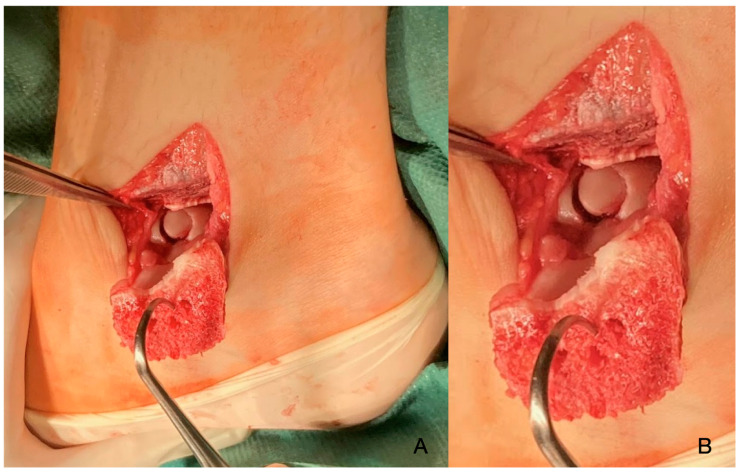
Intra-op images immediately post-osteotomy (**A**), revealing the defect. The close-up image shows well the level of demarcation and the subchondral necrosis (**B**).

**Figure 4 diagnostics-14-01931-f004:**
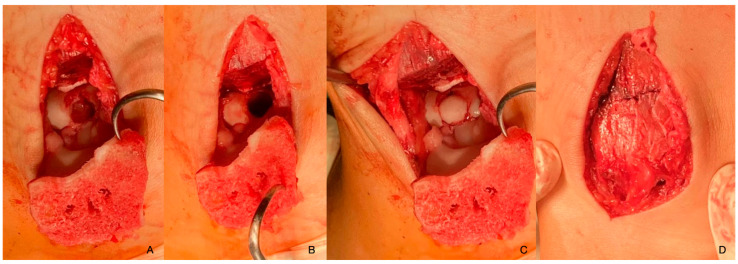
The defect after the removal of the damaged body (**A**,**B**), the articular surface after mosaicplasty (**C**), and the osteotomy fixed with absorbable nails (**D**).

**Figure 5 diagnostics-14-01931-f005:**
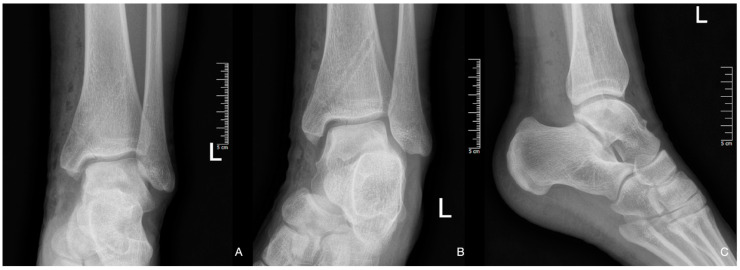
Post-operative X-ray images showing the proper position of the osteotomy with the nail canal visible and the congruent articular surface after mosaicplasty ((**A**): A-P, (**B**): mortise (**C**): Lateral).

**Figure 6 diagnostics-14-01931-f006:**
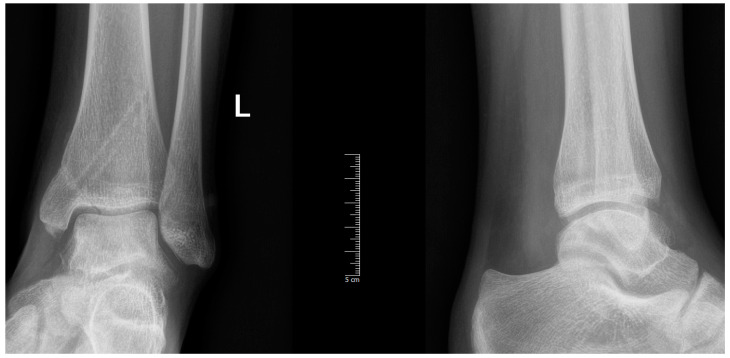
Two-month post-op X-ray, A-P and lateral views.

**Figure 7 diagnostics-14-01931-f007:**
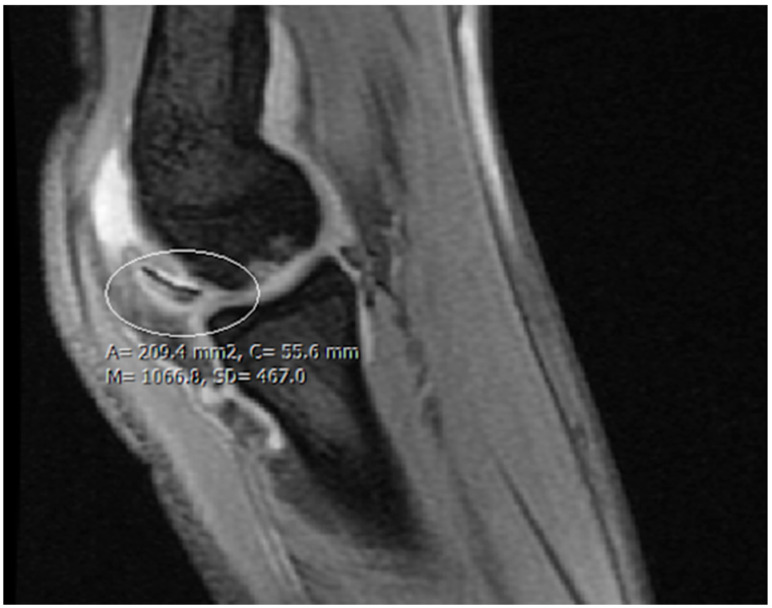
MR image highlighting the chondral piece of the capitulum humeri.

**Figure 8 diagnostics-14-01931-f008:**
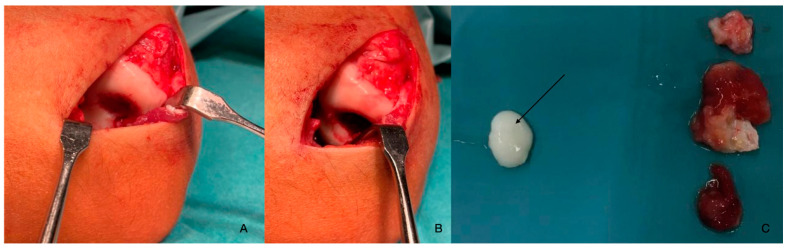
The defect of the humerus (**A**,**B**). The chondral fragment can be seen (black arrow) as well as some synovial coagulum which was removed (**C**).

**Figure 9 diagnostics-14-01931-f009:**
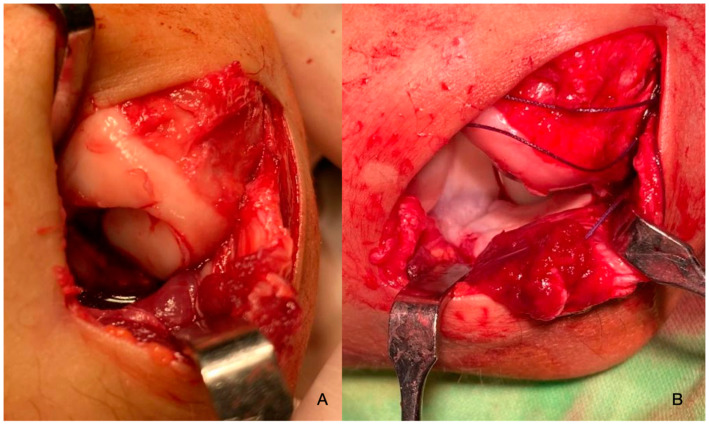
Intraoperative images of the mosaicplasty from anteroradial (**A**) and posterior views (**B**).

**Figure 10 diagnostics-14-01931-f010:**
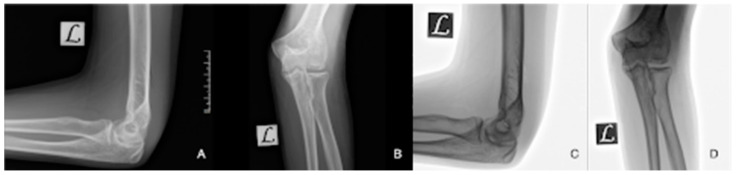
Control X-rays immediately post-op (**A**,**B**). Control X-rays 8 weeks post-op (**C**,**D**).

**Figure 11 diagnostics-14-01931-f011:**
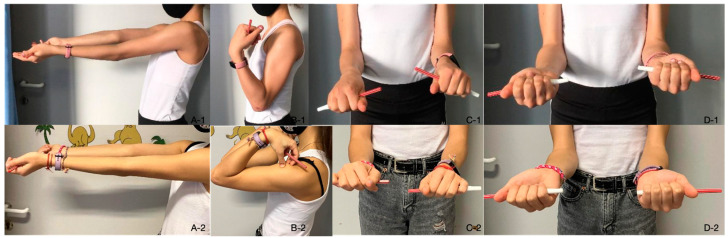
Functional depiction of the elbow joint at 8 weeks (**A-1**–**D-1**), where elbow extension is somewhat limited early on (**A-1**), and full ROM after one year (**A-2**–**D-2**).

**Figure 12 diagnostics-14-01931-f012:**
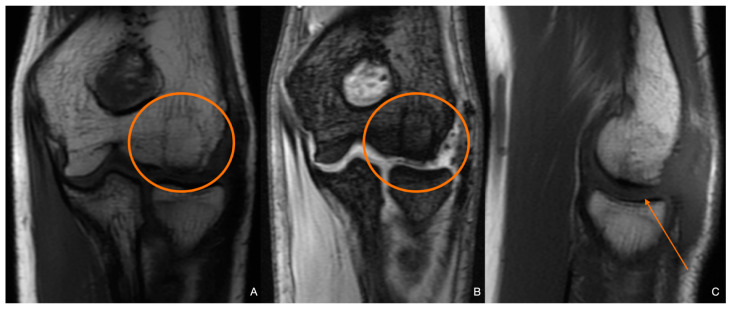
Control MR images six months post-op. The graft can be seen incorporated into the articular surface (orange circles and arrow) on T1 (**A**,**C**)- and T2 (**B**)-weighted images.

**Table 1 diagnostics-14-01931-t001:** Summary of the various types of diagnostic approaches and their corresponding staging which aim to evaluate osteochondral lesions and their characteristic findings.

Imaging:	X-ray	Arthroscopy		MRI		
Classification	Berndt and Harty [21]—1959	Guhl [22]—1982	ICRS by Brittberg and Winalsky [23]—2003	DiPaola [24]—1991	Hefti [4]—1999	Ellerman [15]—2019
Stage 1	Small subchondral compression	Intact Lesion	A stable lesion of the softened area covered by intact cartilage	Thickening of articular cartilage and low signal changes	Small change in signal, without clear fragment margins	Epiphyseal cartilage lesion with necrotic center
Stage 2	Partially detached osteochondral fragment	A lesion with signs of early separation	Lesions with partial discontinuity which are stable when probed	Articular cartilage is breached, with a low signal rim behind the fragment indicating fibrous attachment	Osteochondral fragment with clear margins, without fluid in between	Epiphyseal cartilage lesion with complete or incomplete rim calcification
Stage 3	Completely detached, non-displaced	Partially detached lesion	Lesions with complete discontinuity which are not dislocated (Dead in situ)	High signal changes behind the fragment indicate synovial fluid between the fragment and the underlying subchondral bone	Fluid is partially visible between the fragment and bone	Partially or completely ossified lesion
Stage 4	Completely detached and displaced—loose body	Craters with loose bodies (salvageable or non-salvageable)	Empty defect bed with loose or dislocated fragment	Loose body	Fluid surrounds the fragment but it is still in situ	Healed osseous lesion with scar
Stage 5	Scranton and McDermott modification: Subchondral Cyst	-	-	-	The fragment is completely detached and displaced	Unhealed, detached osseous lesion (Sequestrum)

## Data Availability

All data are contained within the article.

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
