# Peer review of "Juvenile Osteochondritis Dissecans: A Case Report"

_diagnostics, 2024, doi:10.3390/diagnostics14171931_

Round 1

Reviewer 1 Report

Comments and Suggestions for Authors

The case report study provides examples for the diagnosis and treatment guidelines for clinical doctors for OCD. The paper is overall well written. Figures are clear. Following are some comments to improve the manuscript.

Line 39-40, 100.000 should be “100,000”? for incidence?

Line 111,”Trans articular” should be one word “Transarticular”?

Figure 2: Add arrows to point where the OCD are in the images.

Figure 4 legend C “articuéar”. I think this is not an English word.

Some figure panel labels are capital, some are lower case, font sizes are also different. Please correct these.

Line 154, what are the red dots are ? Why the number of weeks was not given?

Line 197 has same issues as line 154. Please address.

Line 207-210, is Figure 12 should be indicated in this paragraph?

Line 245, “Early on graft …..Donor site morbidity” Two incomplete sentences.

Comments on the Quality of English Language

English overall is good. Mino editing is needed.  See comments. 

Reviewer 2 Report

Comments and Suggestions for Authors

This manuscript is presenting two cases of osteochondritis dissecans (OCD) in children. At the first look this is well written manuscript. However, there are multiple areas where improvement could be done.

Firstly, there is no age given in presentation of the case 1 and as well in the case 2. In addition, as journal have name "Diagnostic" those two cases were not diagnostic challenges as they are presented in the manuscript. From my perspective this is something that should be corrected. 

Secondly, authors pointed out that ultrasound could utilized as diagnostic modality in a case of OCD in the lieu of humeral capitellum, but authors did not use ultrasound in the case 2 as diagnostic tool. It is suggested to give one sentence of explanation for this missing diagnostic tool in the case 2.

Third, also, in the line 154 and line 197 there are some important characters missing. See manuscript.

Four, Fig 7 is upside/down as humeral part should be superior in the figure as anatomy suggest.

Five, there is no section on limitations of the study as follow-up is in case 1 one month, only. And in the case 2 one year. In majority of paediatric orthopaedic journals two years follow-up is requested when results of treatment in children are presented.

Six, authors utilize in both cases mosaicplasty as treatment method, however, there is no reference in the list of literature on mosaicplasty. It is suggested to correct what is missing.

Round 2

Reviewer 2 Report

Comments and Suggestions for Authors

Authors have extensively rewritten this two case reports. Reviewers' recommendations and suggestions are well taken.

However, I still do not understand why Fig 7 has lower part of MR image upside/down orientation as well as Fig. 12. When one is observing other figures one can see correct orientation. Would recommend to keep the same direction of figures' orientation through the whole manuscript for the sake of easier reading.  
